# Effectively Prescribing Oral Magnesium Therapy for Hypertension: A Categorized Systematic Review of 49 Clinical Trials

**DOI:** 10.3390/nu13010195

**Published:** 2021-01-10

**Authors:** Andrea Rosanoff, Rebecca B. Costello, Guy H. Johnson

**Affiliations:** 1CMER Center for Magnesium Education &Research, Pahoa, HI 96778, USA; RBCostello@earthlink.net; 2Johnson Nutrition Solutions LLC, Minneapolis, MN 55416, USA; guy@NutritionSolutions.net

**Keywords:** magnesium, oral magnesium therapy, hypertension, blood pressure, anti-hypertensive medications

## Abstract

Trials and meta-analyses of oral magnesium for hypertension show promising but conflicting results. An inclusive collection of 49 oral magnesium for blood pressure (BP) trials were categorized into four groups: (1) Untreated Hypertensives; (2) Uncontrolled Hypertensives; (3) Controlled Hypertensives; (4) Normotensive subjects. Each group was tabulated by ascending magnesium dose. Studies reporting statistically significant (*p* < 0.05) decreases in both systolic BP (SBP) and diastolic BP (DBP) from both baseline and placebo (if reported) were labeled “Decrease”; all others were deemed “No Change.” Results: Studies of Untreated Hypertensives (20 studies) showed BP “Decrease” only when Mg dose was >600 mg/day; <50% of the studies at 120–486 mg Mg/day showed SBP or DBP decreases but not both while others at this Mg dosage showed no change in either BP measure. In contrast, all magnesium doses (240–607 mg/day) showed “Decrease” in 10 studies on Uncontrolled Hypertensives. Controlled Hypertensives, Normotensives and “magnesium-replete” studies showed “No Change” even at high magnesium doses (>600 mg/day). Where magnesium did not lower BP, other cardiovascular risk factors showed improvement. Conclusion: Controlled Hypertensives and Normotensives do not show a BP-lowering effect with oral Mg therapy, but oral magnesium (≥240 mg/day) safely lowers BP in Uncontrolled Hypertensive patients taking antihypertensive medications, while >600 mg/day magnesium is required to safely lower BP in Untreated Hypertensives; <600 mg/day for non-medicated hypertensives may not lower both SBP and DBP but may safely achieve other risk factor improvements without antihypertensive medication side effects.

## 1. Introduction

More than any other modifiable risk factor, hypertension is responsible for cardiovascular disease deaths both globally [1,2] and in the United States [3]. Given the potential harms and costs of hypertension and the need for safe lowering of this important risk factor [4,5,6,7,8], we probed the large trove of research on oral magnesium therapy for hypertension and BP hoping to discern any prescription guidance.

Oral magnesium therapy for the treatment of hypertension has been well studied over the last 35 years but results are highly mixed. The trials differ not only in oral magnesium dose and form of magnesium but also in normotensive vs. hypertensive status at baseline as well as use or non-use of antihypertensive medications. Fourteen clinical trials have shown that oral magnesium therapy significantly lowers both systolic blood pressure (SBP) and diastolic blood pressure (DBP), whereas > twice that number of studies have shown no statistically significant lowering of either SBP, DBP, or both with oral magnesium therapy. Of six meta-analyses on this topic conducted to date [9,10,11,12,13,14], one shows no effect of oral magnesium on BP, one shows lowering of DBP but not SBP, four show that oral magnesium therapy lowers both SBP and DBP but only one of these suggests that the BP reductions are clinically relevant. Such results do not lend confidence in prescribing oral magnesium therapy to control or prevent high blood pressure. However, magnesium’s low cost, safety, positive research in cardiovascular risks [15,16,17] plus its partial beneficial BP results encourages this inclusive analytical categorization of all of these studies. We are looking for information on when and at what dose oral magnesium therapy is beneficial in the treatment of hypertension. This is an inclusive but neither quantitative nor precise analysis, but we hope it will provide guidance for both future meta-analyses and the prescribing of oral Mg therapy for high BP.

## 2. Materials and Methods 

### 2.1. Data Sources and Searches

For this analysis, all articles from six meta-analyses on this subject [9,10,11,12,13,14] and articles used in CMER’s 2016 “Petition for the Authorization of a Qualified Health Claim for Magnesium and Reduced Risk of High Blood Pressure (Hypertension)” [18] submitted to the US Food and Drug Administration (FDA) were added to the CMER collection which began in 1997 (see below and Figure 1). We categorized each of these studies into four groups—Untreated Hypertensive, Uncontrolled Hypertensive, Controlled Hypertensive and Normotensive—and then tabulated each category by ascending oral magnesium dose in milligrams per day. Studies reporting a statistically significant decrease in both SBP and DBP from baseline as well as placebo (if reported) were labeled “Decrease”; all others were deemed “No change” in BP. Our goal was to determine whether these groups might influence the effect of oral magnesium therapy on BP.

### 2.2. Study Selection

The initial 1997 CMER search was performed in PubMed using the following search terms: magnesium AND (blood pressure OR hypertension) [each term limited to Title AND/OR Medical Subject Heading fields]. This initial search yielded 259 studies. A title/abstract scan plus additions from manual searching resulted in 16 studies of oral magnesium therapy for BP for full-text examination and data extraction. These studies were listed by ascending magnesium dose and published in 2003 by Seelig and Rosanoff [19] (non-peer reviewed). CMER continuously updated this preliminary 1997 PubMed search, creating data extraction sheets with data tables for each article and adding appropriate articles cited in published meta-analyses [11,12]. Data sheets for each newly added article were created and updated with correspondence with authors when appropriate. By 2010, the collection had 40 relevant articles that had undergone full-text examination and data extraction. This collection of studies was published, again listed by ascending magnesium dose, after being first categorized as to normotensive (NT) or hypertensive (HT) status as well as medication usage [20] (Figure 1).

In 2017, the 71 articles in four additional published meta-analyses of oral magnesium therapy for BP [9,10,13,14] were added to the CMER collection of 40 articles along with the 45 articles collected for CMER’s 2016 “Petition for a Qualified Health Claim for Magnesium for Hypertension” [18]. This gathering resulted in a total of 156 articles. Duplicates were removed, yielding 58 articles appropriate for an analysis of oral magnesium therapy for BP. CMER then updated its original PubMed search on February 7, 2018, yielding 168 further studies, three of which were appropriate for addition to the collection. All 61 full-text articles plus existing data sheets were gathered for final analysis.

Twelve articles were excluded [21,22,23,24,25,26,27,28,29,30,31,32], yielding 49 articles (see Figure 1). Exclusion criteria were as follows: pregnant subjects [22], use of oral magnesium supplements in combination with any other mineral nutrient [21,23,24], no statistical analysis [25], baseline and/or final SBP or DBP not reported [26,30], a control group using ascorbic acid or imipramine rather than placebo [27,28], only pre- and post-exercise BP values were measured over 2 days [29], and Hypertensive subjects were not separated from Normotensive subjects in the statistical analysis [31,32].

### 2.3. Data Extraction and Quality Assessment

Each article in the final collection of 49 articles was examined for starting SBP/DBP for both magnesium test and placebo control groups to determine BP status at baseline. Subjects with average baseline BP ≥ 140/90 mm Hg or mean blood pressure (MBP) ≥106 mm Hg were deemed hypertensive; all others were deemed normotensive. Studies were furthered examined for antihypertensive medication usage. Antihypertensive medications known to be used in these studies included diuretics (thiazide, spironolactone), calcium channel blockers, beta-blockers, angiotensin-converting enzyme inhibitors, and alpha-blockers.

### 2.4. Data Synthesis and Analysis

Studies were separated into four categories: Untreated Hypertensive, i.e., subjects were treatment naive or not using antihypertensive medications before or during the study and were hypertensive at baseline.Uncontrolled Hypertensive, i.e., subjects were using antihypertensive medications during and previous to the study but were still hypertensive at baseline.Controlled Hypertensive, i.e., subjects were using antihypertensive medications during and previous to the study and were normotensive at baseline.Normotensive subjects, untreated with antihypertensive medications plus normotensive at baseline.

SBP/DBP results with statistical findings and any other cardiovascular-related relevant information were extracted from each article. Studies showing a statistically significant decrease in both SBP and DBP from baseline as well as placebo (if reported) were labeled “Decrease”; all others were deemed “No Change” in BP. Correspondence with authors aided proper categorization as to BP status as well as antihypertensive medication usage.

Each article was examined for oral magnesium form and dose, and each of the four categories was tabulated in order of ascending oral magnesium dose within that category.

### 2.5. Role of the Funding Source

CMER is a group of independent scholars long interested in oral magnesium therapy’s effect on BP [9,10,14,18,20,33,34,35,36,37,38,39]. Funding to sponsor the Food & Drug Administration (FDA)-qualified health claim petition was provided by the Almond Board of California, PepsiCo, Inc., Council for Responsible Nutrition, Pfizer Consumer Healthcare, Premier Magnesia, and Adobe Springs. These sponsors’ donations funded a consultant (Johnson Nutrition Solutions LLC, G.H.J.) to work with CMER scholars to search the literature, evaluate studies by FDA criteria, extract data, and write the FDA petition. The sponsors provided no input into any aspect of this study, data collection, data extraction, writing, or decision to publish. 

## 3. Results

Results for Untreated Hypertensive, Uncontrolled Hypertensive, Controlled Hypertensive and Normotensive subjects are shown in Table 1, Table 2, Table 3 and Table 4, respectively.

### 3.1. Blood Pressure Outcomes with Oral Mg Therapy

In studies on Untreated Hypertensives (Table 1), decrease in BP was highly influenced by magnesium dose: of the 20 studies, only 4 showed “Decrease” in BP by the strict criteria of this analysis, and all of these occurred at daily magnesium doses >600 mg/day. Daily magnesium doses between 120 and 486 mg/day sometimes showed a decrease in either SBP or DBP but not both. The influence of magnesium dose in Untreated Hypertensives is confirmed in the only study using three different magnesium doses in the same subjects (Widman), which showed “No change” at 365 mg Mg/day but “Decrease” at magnesium doses >600 mg/day. It is tempting to predict that Untreated Hypertensive subjects need high doses of oral magnesium (≥600 mg/day) to consistently lower both SBP and DBP, but 2 studies, Walker and Zemel, were both at doses >600 mg/day but showed “No change”. Both of these studies’ authors observed that these subjects were “magnesium replete” either by high measured dietary Mg intake or healthier magnesium-dependent lipoprotein values at baseline, suggesting that the oral Mg therapy >600 mg/day lowers both SBP and DBP in hypertensives but only when subjects are in states of Mg deficit.

In contrast to studies on Untreated Hypertensives, those on Uncontrolled Hypertensive subjects (Table 2) showed that oral magnesium therapy doses as low as 240 mg/day to as high as 607 mg/day consistently and significantly lowered both SBP and DBP (Table 2).

For Controlled Hypertensive subjects (Table 3), Mg doses of 304 to 583 mg/day showed no change in BP in the only two studies of this category.

For Normotensive subjects (Table 4), several studies consistently showed “No change” in BP from Mg doses as low as 250 mg/day and as high as 632 mg/day.

### 3.2. Other Cardiovascular Risk Factors

Even when oral magnesium therapy did not lower BP in Untreated Hypertensives, Controlled Hypertensives or Normotensive subjects, their studies often showed improved parameters linked to cardiovascular health, such as serum or plasma magnesium [40,43,44,46,47,48,74,77,79,80,81,83,84], improved endothelial function [48], reversal of retinal vasospasm [52], improved sodium excretion [43,71], lower sodium in red blood cells [65], higher serum potassium [63], lower C-reactive protein (CRP) [79], improved fasting glucose and insulin resistance [74,78,81,83,88], and lower triglycerides and total cholesterol as well as higher high-density lipoprotein (HDL) cholesterol [74].

### 3.3. Form of Magnesium

Several forms of magnesium, both organic and inorganic, were used in these studies, and it is interesting to note that the only effective doses of >600 mg/day in Untreated Hypertensives were in studies using MgO, often noted in advertising as being poorly absorbed. Magnesium given as aspartate, chloride, oxide, pidolate, lactate, citrate, amino-acid chelate at doses below 600 mg/day were not effective in this category. It was Mg dose, not form of Mg, that made the difference. Likewise, the several forms of magnesium showing BP-lowering effects in Uncontrolled Hypertensives (Table 2) included six forms of magnesium, both organic and inorganic, including MgO, and all were effective in lowering BP by the criteria of this analysis.

### 3.4. Magnesium-Replete Subjects

Two studies on Untreated Hypertensive subjects given 607 and 972 mg Mg/day showed “no change” in BP [54,58]. Walker et al. [54] noted that these subjects were “replete” in magnesium status because their dietary magnesium was high in the magnesium test group (485 vs. 346 mg/day in the placebo group). This author’s term, “magnesium replete” may relate to the intake being >RDA for their Mg test group subjects but <RDA in their placebo group. Zemel et al. [58] noted that their magnesium-treated group was “healthier” than their placebo group at baseline in risk factors affected by magnesium status (see Table 1, footnote 9), and they suggested that oral magnesium “lowers BP only in states of Mg deficiency” (i.e., not magnesium replete). Other studies did not specifically report on the general magnesium status of their subjects, so the question of subjects’ magnesium status playing a role in in magnesium’s effect on BP is not resolved with this analysis even though it is suggested by these two studies.

### 3.5. Treated or Untreated Normotensive Subjects 

None of the studies on subjects normotensive at baseline, be they Controlled Hypertensives or Normotensives (Table 3 and Table 4), showed a decrease in SBP and/or DBP with magnesium doses ranging from 250 to 600 mg/day. In contrast, several but not all studies on subjects hypertensive at baseline, be they Untreated Hypertensives or Uncontrolled Hypertensives (Table 1 and Table 2), showed oral Mg therapy to have a BP-lowering effect.

Tending to confirm this difference between response to oral magnesium therapy between NT and HT subjects, Haga [55] administered 600 mg Mg/day to 17 HT and 8 NT subjects. Only the HT subjects showed a significant decrease in BP. The NT subjects showed no change even at this high level of oral magnesium therapy (see Table 1, footnote 8). In addition, Hattori et al. [69] added a separate analysis of HT versus NT subjects in their magnesium-treated group and found a decrease with the high magnesium dose (607 mg/day) in HT subjects but “no change” in NT subjects (see Table 2, footnote 4).

### 3.6. Side Effects of Oral Magnesium Therapy in These Studies

The trials included in this analysis observed no serious adverse reactions to magnesium supplementation reported among participants receiving up to 972 mg Mg per day. The adverse effects that were reported were minor, transient and were often reported in both experimental and control groups. A full analysis of side effects reported in these trials is available in the Petition to FDA for a Health Claim for Magnesium and Reduced Risk of Hypertension [18] on page 130–132.

### 3.7. Safety of Magnesium Doses in Effective Range

Effective magnesium doses in this analysis ranged from 240 to 972 mg/day. For Untreated Hypertensives, the minimum effective dose was ≥600 mg/day. The tolerable upper intake level (UL) of magnesium for non-food sources is 350 mg/day for adults [15,89]. However, this UL was based on limited data and “although a few studies have noted mild diarrhea and other mild gastrointestinal complaints in a small percentage of patients at levels of 360 to 380 mg per day, it is noteworthy that many other individuals have not encountered such effects even when receiving substantially more than this UL of supplementary magnesium” [18]. Very high intakes of magnesium supplements can be dangerous, even to people without renal or intestinal disease, but such concentrations of magnesium supplement intake are in the range of ≥5000 mg magnesium/d, i.e., ≥10-fold higher than the additional amounts discussed in this article [15,90].

### 3.8. How Does This Analysis Build Upon Existing Meta-analyses?

As noted in the Introduction, six meta-analyses on this topic have been published so far [9,10,11,12,13,14] with mixed results and mostly high heterogeneity. All included randomized trials. This qualitative categorization builds on their quantitative work by suggesting possible origins of that heterogeneity.

One meta-analysis shows no effect of oral magnesium and “high” heterogeneity using 20 trials from all four categories of this collection [12]. A second shows lowering of DBP but not SBP, again with high heterogeneity (I^2^ = 62%/47%), using 12 of the trials from three categories of this collection [11]. Another meta-analysis [10] using 11 trials of only unhealthy subjects drew from three categories of this analysis to show low heterogeneity (I^2^ = 2.1%) in the lowering of both SBP and DBP. Only one published meta-analysis [9] showed zero heterogeneity (I^2^ = 0%) with a clinically relevant lowering of both SBP and DBP (−18.7/−10.9 mm Hg) but using only four trials, all from Group 2. (However, this study’s conclusions are limited by its use of some trials without a placebo control group, as most unbiased data come from subtracting the placebo response from the magnesium BP response so the magnesium effect will not be overestimated in persons with elevated blood pressure at baseline because of regression to the mean.) The two largest and probably most reliable meta-analyses show that oral magnesium therapy lowers both SBP and DBP [13,14] using 33 and 34 trials from all four categories of this analysis, but again with high heterogeneity (I^2^ = 80% + and 62% +).

Combined with our qualitative findings, these results suggest that use of studies on both HT and NT subjects as well as the different effective Mg dose in Untreated Hypertensives vs. Uncontrolled Hypertensive subjects are sources of heterogeneity in Mg for BP meta-analyses.

## 4. Discussion

This categorization clearly shows that NT study subjects, both Controlled Hypertensives and Normotensive (i.e., those with an untreated healthy BP), will not show lower BP with oral magnesium therapy, even at high doses. However, several studies in these normotensive categories reported significant improvement in blood magnesium, lipoproteins, C-reactive protein, fasting glucose and insulin resistance, reversal of retinal vasospasm and increased sodium excretion, all of cardiovascular risk factor benefit. Oral magnesium therapy in NT patients, treated with antihypertensive medications or not, may not show improved BP readings, but these individuals may benefit from improved cardiovascular risk factors.

Among subjects who are hypertensive (≥140/90 mm Hg; MBP ≥ 106 mm Hg) at baseline, both low and high doses of oral magnesium therapy show significant decreases in both SBP and DBP only if the subjects are concurrently taking antihypertensive medications, i.e., partially or Uncontrolled Hypertensives. In the studies of Untreated Hypertensive subjects taking no antihypertensive medications, only the studies with Mg supplement doses >600 mg/day demonstrated statistically significant improvements in blood pressure by the criteria of this analysis. Subjects on lower magnesium doses showed other improvements in measures important to cardiovascular health such as serum magnesium, endothelial function and sodium excretion.

Magnesium-replete subjects, even those who are hypertensive, did not show a decrease in BP with oral magnesium therapy, even at doses as high as 972 mg/day [58]. This finding indicates that a person can have adequate magnesium status and still have high BP. Other essential electrolytes besides magnesium can impact BP. For these patients, potassium could be low, especially when concurrent with a high sodium and/or low calcium intake.

The main limitation to this study is the lack of quantification of the BP changes, instead using the statistics and conclusions from each individual study, which varied widely. This study is not a precise meta-analysis and makes no attempt to fully quantify the impact of the categories derived from this analysis. This, rather, is the job of future meta-analyses, and we see this categorization as a preliminary study to guide future meta-analysis that may provide enhanced information about oral Mg therapy for BP while hopefully achieving lower heterogeneity than existing meta-analyses without losing precision. Nonetheless, this categorization of studies by hypertensive as well as medication status plus magnesium dose yields an informative framework for the prescription of oral magnesium therapy for high BP. It well accommodates large and small studies (*n* = 7–227 receiving magnesium therapy), short-term and long-term studies (2–26 weeks), 11 different forms of magnesium preparations (four inorganic and seven organic), parallel as well as crossover study designs, and placebo control or not (see Michon et al. [62], Sebekova et al. [61], Shafique et al. [60], Motoyama et al. [56], Cohen et al. [52], and Haga [55], which are studies not included in most meta-analyses due to no true placebo group). 

Over 30 years ago, magnesium was shown to alter vascular constriction [91] and several studies have since shown that the physiology and cellular biochemistry of magnesium is important to the functionality of endothelial and smooth muscle cells and regulation of vascular tone [92]. Decreased magnesium concentrations have been implicated in altered vascular reactivity, endothelial dysfunction, vascular inflammation, and structural remodeling [93]. Low dietary magnesium has been associated with a higher risk of hypertension [94]. In the United States, 67% of the population aged ≥51 years is low in dietary magnesium [95] and 55% of adults aged 19 to 50 years, 60% aged 51 to 70 years, and 78% aged >71 years do not consume their estimated average requirement for magnesium [96]. Therefore, it is not surprising that prescribing oral magnesium therapy can lower a high BP. However, this categorized review of clinical trials shows that medication status, hypertensive status, and magnesium dose all must be considered in the use of this inexpensive, non-invasive, safe, readily available, “lifestyle” therapy to prevent and treat high BP as well as other conditions for which high BP is a risk factor. Pervasive low dietary magnesium status affects the health and health care systems of national and global populations [39,97]. Chronic low dietary magnesium quite likely constitutes one of the “lifestyle” components in the high risk of cardiovascular disease of our time [39,98,99].

## 5. Conclusions

This categorization study shows that oral magnesium therapy added to treatment regimens of patients with partially controlled hypertension holds promise as a way of safely achieving lower BP without increasing antihypertensive medications. Prescribing magnesium supplements to hypertensive but untreated patients may not lower BP unless the daily magnesium dose meets or exceeds 600 mg/day, which can be safely and economically accomplished, but magnesium doses below this level can achieve other cardiovascular risk factor improvements without the side effects of antihypertensive medications [99].

## Figures and Tables

**Figure 1 nutrients-13-00195-f001:**
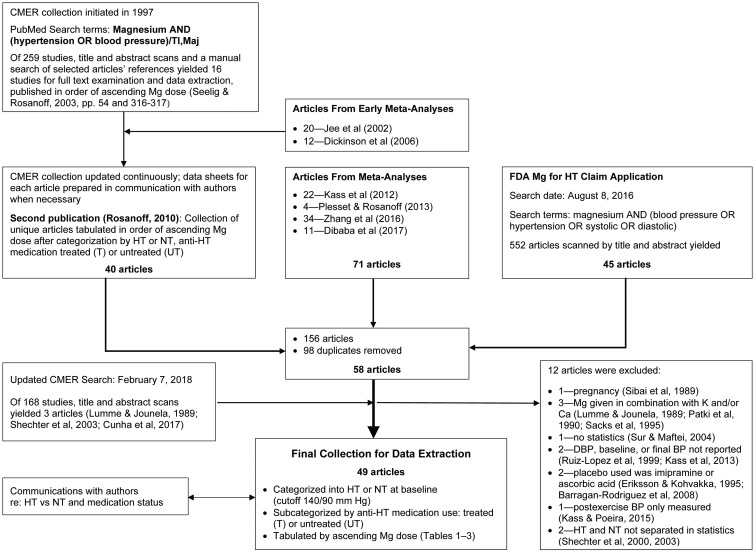
Flow chart of the article acquisition, examination, categorization, and tabulation process. Abbreviations: BP, blood pressure; Ca, calcium; CMER, Center for Magnesium Education & Research; DBP, diastolic blood pressure; HT, hypertensive or hypertension; K, potassium; Maj, Major Topic for National Library of Medicine Pubmed search term; Mg, magnesium; NT, normotensive; T, treated with anti-hypertensive medications; TI, title field for Pubmed search term; UT, untreated, i.e. non-use of anti-hypertensive medications.

**Table 1 nutrients-13-00195-t001:** Summary of magnesium supplementation studies for blood pressure in Untreated Hypertensives (subjects untreated with antihypertensive medications, hypertensive at baseline).

Study Citation	Mg Dose, mg/day	Form of Mg	BP Status at Baseline, NT or HT	Medical Status at Baseline, T or UT	BP Outcome ^1^	Notes
Borrello et al. (1996) [40]	120	MgO	HT	UT	No change ^2^	Decrease in SBP only
Nowson and Morgan (1989) [41]	240	Aspartate	HT	UT	No change	
Ferrara et al. (1992) [42]	365	Pidolate	HT	UT	No change	
Lind et al. (1991) [43]	365	Lactate and citrate	HT	UT	No change ^2,3^	
de Valk et al. (1998) [44]	365	Aspartate HCl	HT	UT	No change ^2^	
Plum-Wirell et al. (1994) [45]	365	Aspartate	HT	UT	No change	
Wirell et al. (1993) [46]	365	Aspartate	HT	UT	No change ^2^	
Cappuccio et al. (1985) [47]	365	Aspartate	HT	UT	No change ^2^	Decrease in DBP only; medication interrupted 2–3 months pre-study
Barbagallo et al. (2010) [48]	368	Pidolate	HT	UT	No change ^2,4^	Baseline BP: 150/82 mm Hg; some perhaps taking medications
Reyes et al. (1984) [49]	384	MgCl_2_	HT	UT	No change	Decrease in DBP only; medication interrupted 4 weeks pre-study
Olhaberry et al. (1987) [50]	384	MgCl_2_	HT	UT	No change	Decrease in SBP only
Purvis et al. (1994) [51]	389	MgCl_2_	HT	UT	No change	Decrease in SBP only
Cohen et al. (1984) [52]	450	MgO	HT	UT	No change ^2,5,6^	
Witteman et al. (1994) [53]	486	Aspartate HCl	HT	UT	No change	Decrease in DBP only
Walker et al. (2002) [54]	607	Amino acid chelate	HT	UT	No change	Mg replete ^7^
Haga (1992) [55]	607	MgO	HT	UT	Decrease ^6^	MBP measured in HT vs. NT “control” ^8^
Motoyama et al. (1989) [56]	607	MgO	HT	UT	Decrease ^6^	No medications during study or 1 month pre-study, at least
Sanjuliani et al. (1996) [57]	607	MgO	HT	UT	Decrease	No medications 2 weeks pre-study or during study
Zemel et al. (1990) [58]	972	Aspartate	HT	UT	No change	Mg replete ^9^; no medications 3 mo pre-study at least
Widman et al. (1993) [59]	365	Mg(OH)_2_	HT	UT	No change	Only titrated Mg dose study
	729	Mg(OH)_2_	HT	UT	Decrease ^10^	
	972	Mg(OH)_2_	HT	UT	Decrease	

Abbreviations: BP, blood pressure; DBP, diastolic blood pressure; HT, hypertensive at baseline; MBP, mean blood pressure; Mg, magnesium; NT, normotensive at baseline; SBP, systolic blood pressure; T, most or all subjects treated with antihypertensive medications including diuretics; UT, most or all subjects treatment naïve or not taking any antihypertensive medications during or before the study. ^1^ Studies showing a statistically significant decrease in both SBP and DBP from baseline as well as placebo (if reported) were labeled “Decrease”; all others were deemed “No change” in BP. ^2^ Rise in serum or plasma Mg in Mg test group. ^3^ Improved sodium excretion in Mg test group. ^4^ Improved endothelial function in Mg test group. ^5^ Reversal of retinal vasospasm. ^6^ Study not included in most meta-analyses due to no true placebo control group. ^7^ Walker et al. [54] showed a very large placebo effect. In addition, this study found dietary Mg especially high (485 mg/day) in the Mg group compared to placebo (346 mg/day) leading authors to suggest those subjects had been “magnesium replete.”. ^8^ Haga [55] gave 600 mg Mg/day to 17 HT and 8 NT “control” subjects. Only HT subjects showed a decrease in MBP. ^9^ Zemel et al. [58] noted that the placebo group at baseline had higher cholesterol, triglycerides, and low-density lipoprotein cholesterol and lower high-density lipoprotein than the Mg group at baseline (i.e., the Mg group was the “healthier” of the two). These differences persisted throughout the study. The authors suggest that oral Mg therapy only lowers BP in “states of Mg deficiency”. ^10^ At the 30 mmol Mg/day dose (729 mg/day), Widman et al. [59] reported both SBP and DBP decreases that were not significant in this crossover design, uncorrected for carryover effects. Statistical analyses were performed on the 30 mmol Mg period against a middle placebo period, which included subjects (50%) who had previously spent 12 weeks taking a 15, 30, and then 40 mmol Mg/day supplement. These statistical analyses did not separate middle placebo group subjects as to pre—or post—Mg arms of the crossover, and neither tested nor corrected for any carryover effect. When a t-test is performed on 30 mmol Mg period BP values against Mg test arm baseline, placebo baseline, and pre-Mg placebo arm final values, both SBP and DBP show significant decreases in all three tests (*p* < 0.01).

**Table 2 nutrients-13-00195-t002:** Summary of magnesium supplementation studies for blood pressure in Uncontrolled Hypertensives (subjects treated with antihypertensive medications, hypertensive at baseline).

Study Citation	Mg Dose, mg/day	Form of Mg	BP Status at Baseline, NT or HT	Medical Status at Baseline, T or UT	BP Outcome ^1^	Notes
Shafique et al. (1993) [60]	240	MgCl_2_	HT	T	Decrease ^2^	Diuretics >1 year
Sebekova et al. (1992) [61]	255	Aspartate HCl	HT	T	Decrease ^2^	Interrupted medications
Michon (2002) [62]	323	Slow-mag/B_6_	HT	T	Decrease ^2^	Beta-blockers, ACE inhibitors, calcium channel blockers, diuretics
Wirell et al. (1994) [63]	365	Aspartate	HT	T	Decrease	Beta-blockers
Dyckner and Wester (1983) [64]	365	Aspartate HCl	HT	T	Decrease	Beta-blockers
Paolisso et al. (1992) [65]	384	Pidolate	HT	T	Decrease ^3^	Thiazide diuretics—long term
Guerrero-Romero and Rodriguez-Moran (2009) [66]	450	MgCl_2_	HT	T	Decrease	All taking medications ≥6 months pre-study, type not specified
Kawano et al. (1998) [67]	486	MgO	HT	T	Decrease	33% untreated; 30% monotherapy; 37% combination therapy; therapy included calcium channel blockers, beta-blockers, ACE inhibitors, thiazides, spironolactone, alpha-blockers
Cunha et al. (2017) [68]	600	Mg chelate	HT	T	Decrease	Hydrochlorothiazide
Hattori et al. (1988) [69]	607	MgO	HT	T	Decrease ^4^	Thiazide diuretics—long term
			NT	T	No change ^4^	Thiazide diuretics—long term

Abbreviations: ACE, angiotensin-converting enzyme; BP, blood pressure; DBP, diastolic blood pressure; HT, hypertensive at baseline; MBP, mean blood pressure; Mg, magnesium; NT, normotensive at baseline; SBP, systolic blood pressure; T, most or all subjects treated with antihypertensive medications including diuretics, ACE inhibitors, calcium channel blockers, beta-blockers, or alpha-blockers. ^1^ Studies showing a statistically significant decrease in both SBP and DBP from baseline as well as placebo (if reported) were labeled “Decrease”; all others were deemed “No change” in BP. ^2^ Study not included in most meta-analyses due to no true placebo control group. ^3^ Mg test group showed lower sodium in red blood cells. ^4^ Hattori et al. [69] showed significant decreases in both SBP and DBP from baseline and placebo in these 20 thiazide-treated subjects. However, a baseline BP of 134/80 mm Hg would categorize them as NT. However, the authors separated the 9 HT subjects (baseline MBP = 104.8 mm Hg) from the 11 NT subjects (baseline MBP = 93) and found that the former showed a decrease in MBP (−11 ± 2.0 mm Hg, *p* < 0.05) and the latter showed no change in MBP (+ 0.1 ± 0.46 mm Hg).

**Table 3 nutrients-13-00195-t003:** Summary of magnesium supplementation studies for blood pressure in Controlled Hypertensives (subjects treated with antihypertensive medications, normotensive at baseline).

Study Citation	Mg Dose, mg/day	Form of Mg	BP Status at Baseline, NT or HT	Medical Status at Baseline, T or UT	BP Outcome ^1^	Notes
Henderson et al. (1986) [70]	304	MgO	NT	T	No change	Potassium depleting diuretics ≥ 6 months
Itoh et al. (1997) [71]	413–583	Mg(OH)2	NT	T and UT	No change ^2,3^	Some subjects were borderline HT; medications kept constant “when necessary” (medications not specified)

^1^ Studies showing a statistically significant decrease in both SBP and DBP from baseline as well as placebo (if reported) were labeled “Decrease”; all others were deemed “No change” in BP. ^2^ Rise in Na excretion; decrease in serum Na. ^3^ Faulty baseline statistics; final SBP and DBP significantly lower than baseline but change in SBP and DBP not significantly different from those of placebo, thus the “No change” categorization. Only as a percentage of run-in, pre-baseline value was final Mg SBP significantly lower than placebo’s percentage of run-in SBP. Abbreviations: BP, blood pressure, NT, normotensive at baseline; HT, hypertensive at baseline; UT, most or all subjects treatment naïve or not taking any antihypertensive medications during or before the study; T, most or all subjects treated with antihypertensive medications including diuretics.

**Table 4 nutrients-13-00195-t004:** Summary of magnesium supplementation studies for blood pressure in Normotensives (subjects untreated with antihypertensive medications, normotensive at baseline).

Study Citation	Mg Dose, mg/day	Form of Mg	BP Status at Baseline, NT or HT	Medical Status at Baseline, T or UT	BP Outcome ^1^	Notes
Doyle et al. (1999) [72]	250	Mg(OH)_2_	NT	UT	No change	
Lee et al. (2009) [73]	300	MgO	NT	Unknown	No change	
Guerrero-Romero et al. (2004) [74]	304	MgCl_2_	NT	UT	No change	
Sacks et al. (1998) [75]	340	Lactate	NT	UT	No change	
Joris et al. (2016) [76]	350	Citrate	NT	UT	No change	Overweight, healthy
TOHP Study Group (1992) [77]	365	Diglycine	NT	UT	No change	
Mooren et al. (2011) [78]	365	Aspartate HCl	NT	Not reported	No change	
Simental-Mendia et al. (2014) [79]	382	MgCl_2_	NT	UT	No change	
Simental-Mendia et al. (2012) [80]	382	MgCl_2_	NT	UT	No change	
Rodriguez-Moran and Guerrero-Romero (2014) [81]	381	MgCl_2_	NT	UT	No change ^2^	Hyperglycemic, insulin resistant, hypertriglyceridemic, hypomagnesemic, normal weight
Cosaro et al. (2014) [82]	394	Pidolate	NT	UT	No change	
Rodriguez-Moran and Guerrero-Romero (2003) [83]	450	MgCl_2_	Borderline HT/NT	UT	No change	
Rodriguez-Hernandez et al. (2010) [84]	450	MgCl_2_	NT	UT	No change	
Daly et al. (1990) [85]	500	MgO	NT	UT	No change	
Kisters et al. (1993) [86]	505	Aspartate	NT	UT	No change	
Wary et al. (1999) [87]	600	Lactate + B_6_	NT	UT	No change	
Guerrero-Romero and Rodriguez-Moran (2011) [88]	632	MgCl_2_	NT	UT	No change ^3^	Subjects had low serum Mg that normalized with Mg therapy

Abbreviations: BP, blood pressure; DBP, diastolic blood pressure; HOMA-IR, homeostatic model assessment–insulin resistance; HT, hypertensive at baseline; MBP, mean blood pressure; Mg, magnesium; NT, normotensive at baseline; SBP, systolic blood pressure; T, most or all subjects treated with antihypertensive medications including diuretics; UT, most or all subjects treatment naïve or not taking any antihypertensive medications during or before the study. ^1^ Studies showing a statistically significant decrease in both SBP and DBP from baseline as well as placebo (if reported) were labeled “Decrease”; all others were deemed “No change” in BP. ^2^ SBP and DBP in the Mg group showed no statistically significant change from baseline, but both decreased significantly compared with the placebo group (which showed slight increases in both SBP and DBP), thus the “no change” categorization. Nonetheless, subjects taking Mg significantly improved fasting glucose, HOMA-IR index, triglycerides, and serum Mg when statistically compared with both baseline and placebo. ^3^ Author statistics showed baseline Mg group vs. placebo SBP and DBP to be not significant and final Mg group vs. placebo SBP and DBP to be significantly different (*p* < 0.05). They did not calculate *p* for final vs. baseline. In our calculations, the placebo group showed no change in both DBP and SBP from baseline; Mg test group final vs. baseline for SBP was highly significant (*p* = 0.0003) but borderline for DBP (*p* = 0.0642) which technically requires a “No change” by criteria of this analysis. Any reasonable person would deem this a “Decrease” which a quantitative meta-analysis would incorporate.

## Data Availability

Data sharing not applicable.

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
