# Peer review of "Effectively Prescribing Oral Magnesium Therapy for Hypertension: A Categorized Systematic Review of 49 Clinical Trials"

_nutrients, 2021, doi:10.3390/nu13010195_

Round 1

Reviewer 1 Report

The manuscript is a systematic review of oral magnesium supplementation in persons with elevated blood pressure, both treated and treated, and normotensive persons which include both hypertensive patients with BP < 140/90 and non-hypertensive persons with BP < 140/90.  There are several major concerns with the current presentation of the data that limit its utility to the clinical an scientific community:

1) there is no quantification of the impact of magnesium on lowering of blood pressure as all studies with a reduction attributable to magnesium were simply put in the "decrease" group if both systolic and diastolic BP were lowered.  The absence of this quantification severely limits the utility of this data.

2) the studies were characterized by BP level and secondarily by whether they were receiving treatment or not.  This leads to heterogenous groups.  For example, normotensive subjects could be those who are treated with antihypertensive medication but have BP < 140/90 mm Hg or those who are truly normotensive.  The hypertensive treated and un-treated groups all had BP >= 140/90 and are therefore not just hypertensive but more specifically they are uncontrolled hypertensives.  The way these groups are labeled should be adjusted to be more specific

3) the reader should be given the rationale for separating the groups primarily on whether BP is uncontrolled as opposed to using a traditional definition of hypertension which would remove the treated hypertensives with controlled BP out of the normotensive group.  

4) in the data synthesis and analysis section (2.4), it is not clear as to whether studies included in this analysis had to have a control group or not.  By far, the most unbiased data, comes from subtracting the placebo response from the magnesium BP response.  Otherwise, the magnesium effect will be overestimated in persons with elevated blood pressure at baseline because of regression to the mean.  The only credible way to present this data is to use placebo-adjusted magnesium responses and the only way to do this is to ensure that the only studies included in this analysis had a placebo control group.  It is irrelevant how much the BP changed from baseline in the magnesium or control group, it is highly relevant though what the placebo-adjusted BP change is.  An example of the ambiguity of what BP change is being expressed in the manuscript is found on line 114 and 115 and line 143 and 144. 

5) elevated BP should be consistently defined as >= 140/90 or MAP >= 106 mm Hg.  In some areas elevated BP is defined as > 140/90 or MAP > 106.  Please correct.  

6) In the abstract and also in the manuscript the term "magnesium replete" is used yet there is no definitive criterion upon which this is based.  there is an allusion to the notion that this may have to do with dietary intake being adequate but for sure such data would only be found in a research setting where dietary histories are available.  It should be clarified as to what the definition of magnesium replete is and to how many studies magnesium status was available for.

7) on page 7, line 65, showing BP change at decimal places is excessive; one decimal point is enough.  

8) page 11, lines 189 - 190, it is unclear what this statement means "Again, subjects with magnesium levels to low to show BP decreases...."

Author Response

Reviewer 1 for Nutrients

The manuscript is a systematic review of oral magnesium supplementation in persons with elevated blood pressure, both treated and treated, and normotensive persons which include both hypertensive patients with BP < 140/90 and non-hypertensive persons with BP < 140/90.  There are several major concerns with the current presentation of the data that limit its utility to the clinical an scientific community:

1) there is no quantification of the impact of magnesium on lowering of blood pressure as all studies with a reduction attributable to magnesium were simply put in the "decrease" group if both systolic and diastolic BP were lowered.  The absence of this quantification severely limits the utility of this data.  Research trials on this subject have varied widely, even the 6 meta-analyses, five of which show high heterogeneity.  The goal of this categorization is a preliminary analysis meant to provide prescription guidance and suggest categories for a future meta-analysis that might lower heterogeneity found in most existing meta-analyses.  Description and discussion of these aspects have been incorporated in the editing and revision of Abstract, Introduction and specifically addressed in the Discussion in paragraph #4 of Discussion on page 12.  (Sorry, line numbers are all changing on me and I don’t want to confuse.)

2) the studies were characterized by BP level and secondarily by whether they were receiving tr1atment or not.  This leads to heterogenous groups.  For example, normotensive subjects could be those who are treated with antihypertensive medication but have BP < 140/90 mm Hg or those who are truly normotensive.  The hypertensive treated and un-treated groups all had BP >= 140/90 and are therefore not just hypertensive but more specifically they are uncontrolled hypertensives.  The way these groups are labeled should be adjusted to be more specific  We have revised and renamed the categories to “Uncontrolled Hypertensives” now in Table 2, “Untreated Hypertensives” now in Table 2, added Table 3 for “Controlled Hypertensives”  (i.e. the treated subjects who were normotensive at baseline) and “Normotensives” in the newly created Table 4.   We appreciate the better layout of information this suggestion provides, and have revised and edited the text to show this better categorization in the Abstract, Introduction, Data Synthesis and Analysis, The Tables themselves, Results on Blood Pressure Outcomes With Oral Mg Therapy, and in the Discussion.

3) the reader should be given the rationale for separating the groups primarily on whether BP is uncontrolled as opposed to using a traditional definition of hypertension which would remove the treated hypertensives with controlled BP out of the normotensive group.  The renaming of categories plus creation of the new Table 3 category of Controlled Hypertensivesas separate from Normotensives in the new Table 4 takes care of this proper concern.  We have reviewed ad clearly noted the definition of hypertension in all trials tabulated in this analysis, i.e. >140/90.   

4) in the data synthesis and analysis section (2.4), it is not clear as to whether studies included in this analysis had to have a control group or not.  Each trial not having a true placebo or control group is now marked via footnote in appropriate tables with the term “Study not included in most meta-analyses due to no true placebo control group”.  By far, the most unbiased data, comes from subtracting the placebo response from the magnesium BP response.  Otherwise, the magnesium effect will be overestimated in persons with elevated blood pressure at baseline because of regression to the mean.  The only credible way to present this data is to use placebo-adjusted magnesium responses and the only way to do this is to ensure that the only studies included in this analysis had a placebo control group. In the Results section on “How Does This Analysis Build Upon Existing Meta-analyses?” this point is explicitly made about half way through the first paragraph on pg 12.  This point has been kept in mind during all revision and edits.  See specifically footnote 3 of Table 4.    It is irrelevant how much the BP changed from baseline in the magnesium or control group, it is highly relevant though what the placebo-adjusted BP change is. We agree, and plan a quantitative meta-analysis that will use placebo-adjusted BP changes and the framework provided by this preliminary study’s categories.

Results of statistical tests presented in the Tables were all reported by original trial authors. In two trials, additional statistics were added to those reported as marked in Table 1 (Widman) and Table 4 (Guerrero-Romero & Rodriguez-Moran, 2011) footnotes.  Most trials in this analysis had placebo control groups and most of which did subtract placebo change values from Mg test group values, so statistics on these is what was used to designate “Decrease” or “No change” status.   So the reader of this paper can discern, we have marked the trials in table footnotes that did not have a placebo or control group.      An example of the ambiguity of what BP change is being expressed in the manuscript is found on line 114 and 115 and line 143 and 144.   We agree.   These points of text were deleted during manuscript editing and revision process to better reflect the renaming and categorization of BP categories as well as emphasizing the qualitative nature of this analysis.

5) elevated BP should be consistently defined as >= 140/90 or MAP >= 106 mm Hg.  In some areas elevated BP is defined as > 140/90 or MAP > 106.  Please correct.  Done as requested throughout manuscript.

6) In the abstract and also in the manuscript the term "magnesium replete" is used yet there is no definitive criterion upon which this is based.  there is an allusion to the notion that this may have to do with dietary intake being adequate but for sure such data would only be found in a research setting where dietary histories are available.  It should be clarified as to what the definition of magnesium replete is and to how many studies magnesium status was available for.   More thorough discussion of this topic has been added to Section 3.4.  See added wording in red.    

7) on page 7, line 65, showing BP change at decimal places is excessive; one decimal point is enough.  Corrected.

8) page 11, lines 189 - 190, it is unclear what this statement means "Again, subjects with magnesium levels to low to show BP decreases...."  Agreed.  This has been corrected in revision and editing in the Second paragraph of Discussion on pg 12.  See added wording in red.    

Reviewer 2 Report

Rosanoff et al report a systematic review of 49 clinical trials on oral Mg for blood pressure lowering in hypertensives and normotensives separately. Based on qualitative summary, studies suggest Mg supp may be useful for treated and untreated hypertensives and normotensives. The paper is generally well written, but I have a number of major concerns.

  1. How does this systematic review on Mg supplementation for hypertension build on the 6 existing meta-analyses that examine Mg therapy for BP reductions?
  2. The authors mention the lower hypertension threshold to SBP>=130 mmhg and DPG=80 mmhg? Were any of the studies included conducted after the new AHA guidelines? It wasn’t clear if/how this was addressed during the review process. This seems like a way to set the review apart from prior meta-analyses.
  3. Did the prior meta-analyses and the eligible studies considered only include randomized controlled trials? This could be stated explicitly within the methods.
  4. Lines 109-112 – This footnote highlights the challenge with lumping SBP and DBP as one BP outcome. Perhaps a second column separating out SBP and DBP would help avoid the classification from being “oddly wrong”.
  5. Were the other cardiovascular risk factors an outcome that was specified a priori or ad hoc? These aspects seem tangential to the main aims of the paper. was urinary magnesium studies specifically sought? Or was urinary Mg only of interest if it happened to be reported in one of the 49 studies included in the review?
  6. The authors mention strengths of the findings. Could the authors also comment on potential limitations of the search strategy and/or drawing inferences from the trials found (heterogeneity in treatments, age, formulation etc)? Does any formulation of supplement containing magnesium >600 mg/day help or are specific Mg forms preferable?
  7. While I agree that Mg supp can be safe (particularly vs antihypertensive medications), the systematic review did not examine safety and the findings don’t look at MG supp safety. Potential for GI side effects (b/c Mg can act similar to a laxative) should be mentioned. This side effect while usually mild (if it occurs)? Moreover during the course of full text review, did any of the trials report any side effects related the intervention? Is a threshold at which Mg supplementation could result in harm (in extreme doses that could result in for example arrhythmias)? Touching on the “there can be too much of a good thing” would help balance the findings.
  8. the conclusion about certain doses being ‘required’ do not seem supported. For example, Line 125 “HT-UT subjects needed higher doses of oral magnesium”. This wording makes it sound causal as though the same subjects participated in all the studies included with all the same doses/formulations for Mg and antihypertensive use, but required a high dose of oral Mg. For example, I suggest rephrasing to (and applicable to other places in the text) something like : “In the studies of HT-UT subjects, only the studies with Mg supp doses >=600 mg/day demonstrated statistically sig improvements in blood pressure.”

Minor

  1. Abstract - “Can magnesium dose and medication status guide prescribing?” This sentence seems out of place.
  2. Line 38 – what is a “true lowering” of SBP, DBP?
  3. Henderson et al. (1986) [70] - why was this group without hypertension at baseline being treated with antihypertensives? Is this study truly normotensive at baseline or are they treated hypertensives?
  4. was the statistical significance threshold p<0.05?
  5. How did the prior studies define ‘magnesium replete’?

Author Response

Reviewer 2 comments – Nutrients

Rosanoff et al report a systematic review of 49 clinical trials on oral Mg for blood pressure lowering in hypertensives and normotensives separately. Based on qualitative summary, studies suggest Mg supp may be useful for treated and untreated hypertensives and normotensives. The paper is generally well written, but I have a number of major concerns.

  1. How does this systematic review on Mg supplementation for hypertension build on the 6 existing meta-analyses that examine Mg therapy for BP reductions?  This topic is now discussed in new section added at on Pg 12, “How Does This Analysis Build Upon Existing Meta-analyses?”  
  2. The authors mention the lower hypertension threshold to SBP>=130 mmhg and DPG=80 mmhg? Were any of the studies included conducted after the new AHA guidelines? It wasn’t clear if/how this was addressed during the review process. This seems like a way to set the review apart from prior meta-analyses.  This study used only the 140/90 as definition of HT vs NT.  Text has been rewritten to emphasize this point, and text referring to other guidelines have been deleted so as not to confuse.  Whelton et al., 2017 reference on new guidelines has been retained and loosely mentioned in Conclusion.
  3. Did the prior meta-analyses and the eligible studies considered only include randomized controlled trials? This could be stated explicitly within the methods.  This topic is covered in the newly added section 3.8, “How Does This Analysis Bild Upon Existing Meta-analyses?” on pg 12.  Topic is specifically covered in second sentence of the paragraph and about half way downin discussion of the one trial that included non-controlled trials (sorry, the line numbers keep changing on me). In addition each trial without a control or true placebo has been so marked in the Tables.  See Footnote 6 for Table 1 and Footnote 2 for Table 2.
  4. Lines 109-112 – This footnote highlights the challenge with lumping SBP and DBP as one BP outcome. Perhaps a second column separating out SBP and DBP would help avoid the classification from being “oddly wrong”.   This study was changed to the properly designated “No change” by the criteria of this analysis, but more information was added in the footnote 3 of Table 4.  The revisions and edits emphasize more that quantitative meta-analysis of these trials is needed to actually quantify BP results with Mg therapy.  This categorized analysis is only for the purpose of prescription guidance and, hopefully, helping discern underlying sources of heterogeneity in existing meta-analyses, as discussed in section 3.8.
  5. Were the other cardiovascular risk factors an outcome that was specified a priori or ad hoc? Studies varied in their goals, some only measuring BP, others including particular health categories, i.e. diabetes, etc.   Mg supplement trials showing BP outcome were sought, and if other cardiovascular risk factors were also measured, these results are now reported with footnotes in the tables because their outcome is important to the main aims of this paper.  These aspects seem tangential to the main aims of the paper. was urinary magnesium studies specifically sought? No.  Or was urinary Mg only of interest if it happened to be reported in one of the 49 studies included in the review?  The latter.  Trials showing urinary Mg changes are no longer marked nor listed as that measure is not really associated with cardiovascular risk as the other factors mentioned and listed.
  6. The authors mention strengths of the findings. Could the authors also comment on potential limitations of the search strategy and/or drawing inferences from the trials found (heterogeneity in treatments, age, formulation etc)?The limitations of this study is now included in the 4th paragraph of the Discussion.    Does any formulation of supplement containing magnesium >600 mg/day help or are specific Mg forms preferable?  Discussion of forms of magnesium in these trials is not discussed in Section 3.3 of Results. 
  7. While I agree that Mg supp can be safe (particularly vs antihypertensive medications), the systematic review did not examine safety and the findings don’t look at MG  supp safety.  Safety discussion of oral Mg therapy added at Section 3.6 and 3.7 of Results.              Potential for GI side effects (b/c Mg can act similar to a laxative) should be mentioned. This side effect while usually mild (if it occurs)? Moreover during the course of full text review, did any of the trials report any side effects related the intervention?  Side effects of all studies was thoroughly studied in the preparation of the Petition to FDA (ref 15) and is now summarized in Section 3.6 of Results.          .  Is a threshold at which Mg supplementation could result in harm (in extreme doses that could result in for example arrhythmias)? Touching on the “there can be too much of a good thing” would help balance the findings.  Discussion of DRI UL is now included in Section 3.7 and both upper limit set by DRI as well as actual toxic dose is shown with references.
  8. the conclusion about certain doses being ‘required’ do not seem supported. For example, Line 125 “HT-UT subjects needed higher doses of oral magnesium”. This wording makes it sound causal as though the same subjects participated in all the studies included with all the same doses/formulations for Mg and antihypertensive use, but required a high dose of oral Mg. For example, I suggest rephrasing to (and applicable to other places in the text) something like : “In the studies of HT-UT subjects, only the studies with Mg supp doses >=600 mg/day demonstrated statistically sig improvements in blood pressure.”   Suggestion is appreciated and improved wording and concept used throughout manuscript editing and revision.

Minor

  1. Abstract - “Can magnesium dose and medication status guide prescribing?” This sentence seems out of place.  Deleted and abstract rewritten to better reflect revisions.
  2. Line 38 – what is a “true lowering” of SBP, DBP?  Changed to “statistically significant lowering” at line 42.
  3. Henderson et al. (1986) [70] - why was this group without hypertension at baseline being treated with antihypertensives? Is this study truly normotensive at baseline or are they treated hypertensives?  Good point.   This study was moved into the new category, “Controlled Hypertensives”, i.e Table 3.  This newly created table includes studies where subjects are treated with antihypertensive meds and have BP <140/90 at baseline.
  4. was the statistical significance threshold p<0.05?  yes – This is the level of significance cchosen by all trial authors, and their statistics are what have been used in this analysis.  In two cases, additional statistics shed ore light, and these are marked in Table 1 and 4 footnores. 
  5. How did the prior studies define ‘magnesium replete’?  Discussion added at section 3.4 of Results. 

Submission Date

Round 2

Reviewer 2 Report

The authors addressed my comments satisfactorily and the paper is improved.

Author Response

Thank you for your comments in Round one, and two, they helped make this a better paper.